# Equivariant Reinforcement Learning under Partial Observability

**Hai Nguyen, Andrea Baisero, David Klee, Dian Wang, Robert Platt, Christopher Amato**
Khoury College of Computer Sciences, Northeastern University, Boston, MA, United States
`nguyen.hai1@northeastern.edu`
`https://sites.google.com/view/equi-rl-pomdp`

**Abstract:** Incorporating inductive biases is a promising approach for tackling challenging robot learning domains with sample-efficient solutions. This paper identifies partially observable domains where symmetries can be a useful inductive bias for efficient learning. Specifically, by encoding the equivariance regarding specific group symmetries into the neural networks, our actor-critic reinforcement learning agents can reuse solutions in the past for related scenarios. Consequently, our equivariant agents outperform non-equivariant approaches significantly in terms of sample efficiency and final performance, demonstrated through experiments on a range of robotic tasks in simulation and real hardware.

**Keywords:** Partial Observability, Equivariant Learning, Symmetry

## 1 Introduction

A key challenge in robot learning is to improve sample efficiency, i.e., to reduce the number of experiences or demonstrations needed to learn a good policy. One way to do this is to identify domain symmetries that can structure the policy space. Recent works have demonstrated that symmetry-preserving (equivariant) neural network models are a particularly effective way of accomplishing this [1, 2, 3, 4, 5]. However, these works have focused primarily on fully observable Markov decision processes (MDPs) rather than partially observable systems encoded as partially observable MDPs (POMDPs) [6]. The question arises whether symmetric neural models can also be used to solve Partially Observable Reinforcement Learning (PORL) problems. This paper identifies the theoretical conditions under which this is indeed the case and describes an equivariant recurrent model that works well in practice.

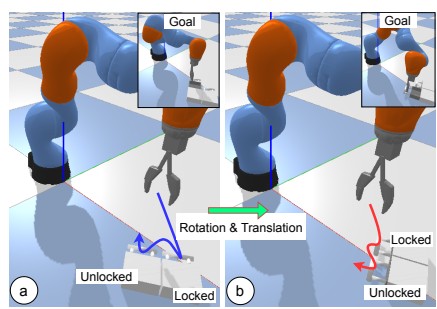

Figure 1: `Drawer-Opening`: This POMDP is rotationally symmetric in the sense that an optimal solution to the problem on the left (in blue) can be rotated to obtain an optimal solution to a rotated version of the problem on the right (in red).

To motivate, Fig. 1 illustrates the `Drawer-Opening` problem where a robot is presented with a chest containing two drawers, one locked and one unlocked. To solve this task, the robot must determine which drawer is unlocked and then open that drawer, relying only on top-down image observations. This task reflects a common POMDP when physical properties (whether a drawer is unlocked) are hidden from the visual input. The only way for the robot to distinguish between the two drawers is to attempt to open one of them. This is a classic feature of a POMDP – that the agent must perform *information gathering actions* to obtain information needed to solve the task. Notice that this problem is rotationally symmetric in the sense that its optimal solution (the blue end-effector trajectory in Fig. 1a) rotates (the red trajectory in Fig. 1b) when the scene itself rotates and is an example of the type of symmetry that we want our agents to embed in their architectures.

We make three contributions in this work. First, we extend the framework of group-invariant Markov decision processes [1] to the partially observable setting, resulting in a new theory and solution

7th Conference on Robot Learning (CoRL 2023), Atlanta, USA.

method. Specifically, we prove the optimal policy and the value function must be equivariant and invariant in this new setting. Second, backed by the proof, we introduce equivariant actor-critic agents that inherently embed the domain symmetry in their architectures. Finally, we apply the agents in realistic robot manipulation tasks with sparse rewards, where our agents are shown to significantly outperform non-equivariant approaches in both sample efficiency and final performance. Our approach's effectiveness is shown through simulated and real-robot experiments with equivariant and recurrent versions of Advantage Actor-Critic (A2C) [7] and Soft Actor-Critic (SAC) [8].

## 2 Related Works

**Learning under Partial Observability** Unlike classical planning-based methods [9, 10, 11] that impractically require the complete dynamics of the environment, learning-based methods [12, 13, 14, 15, 16, 17, 18, 19] utilize recurrent versions of common reinforcement learning (RL) algorithms for policy learning by directly interacting with the environment. To speed up learning, some methods leverage privileged information assumed available during training, such as the states, the belief about the environment states, or the fully observable policy [20, 21, 22, 23], which are orthogonal to our approach. Only a few prior works exploited domain symmetries under partial observability. Kang and Kim [24], Doshi and Roy [25] leveraged the invariance of the value function of some POMDPs given a state permutation and experimented on a classical planning-based method [11] with the above limitations. Recently, Muglich et al. [26] used equivariant networks to enforce symmetry when multiple agents coordinate. In contrast, we use model-free RL agents in a single-agent setting.

**Equivariant Learning** Equivariant networks have been successfully applied to a range of tasks such as point cloud analysis [27] and molecular dynamics [28, 29]. A common approach is to build networks with group equivariant convolutions [30] which are equivariant to arbitrary symmetry groups, such as 2D [31, 32] and 3D [27, 33, 34, 35] transformations. Recently, for MDPs, equivariant networks have been applied to robotics [2, 1, 5] and reinforcement learning [36, 37] to improve sample efficiency. Closest to our work is [1], which formalized group-invariant MDPs and used equivariant networks to perform robotic manipulation tasks. In contrast, this work extends equivariant reinforcement learning to partially observable environments, resulting in a new theory and method.

**Equivariance v.s. Data Augmentation** Both methods leverage known domain symmetry to improve learning, but in different ways. On the one hand, data augmentation artificially expands the training data distribution with transformed versions of the data using the symmetry (e.g., rotating, cropping, or translating images [38, 39]); then training a non-equivariant model. On the other hand, an equivariant approach bakes the domain symmetry directly into the model's weights, so an equivariant model can automatically generalize across input transformations even before training. Compared to an equivariant approach, a model trained using data augmentation alone is often less sample efficient [32, 34], generalizes worse [40], and requires a bigger architecture and longer training time for the same performance due to the extra work of learning symmetry injected in the data.

## 3 Background

Here, we review some background about POMDPs, some specific group theories used in our work, and finally, the basis of our approach — the framework of group-invariant MDPs [1].

### 3.1 Partially Observable Markov Decision Processes

A POMDP is defined by a tuple $(\mathcal{S}, \mathcal{A}, \Omega, b_0, T, R, O)$, where $\mathcal{S}$, $\mathcal{A}$, and $\Omega$ are the state space, the action space, and the observation space, respectively. $b_0 \in \Delta\mathcal{S}$ is the starting state distribution (a.k.a. the initial belief), states change and observations are emitted according to the stochastic dynamics function $T(s, a, s')$ and the stochastic observation function $O(a, s', o)$, respectively. Generally, an optimal agent may need to choose actions based on the entire observable action-observation history $h_t = (o_0, a_0, \ldots, a_{t-1}, o_t)$ [41]. Denoting the space of all histories as $\mathcal{H}$, the goal is to find a history-policy $\pi \colon \mathcal{H} \to \Delta\mathcal{A}$ which maximizes the expected discounted return $J = \mathbb{E}\left[\sum_{t=0}^{\infty} \gamma^t R(s_t, a_t)\right]$, where $\gamma \in [0, 1)$ is a discounting factor. An important concept in POMDPs is the belief $b(s) = \Pr(s \mid h)$, which is the probability that the true state is $s$ given an observed history $h$. The belief state is a sufficient statistic of the history, sufficient for optimal control. However, updating the belief state requires complete knowledge of the POMDP dynamic models, which are often hard to obtain.

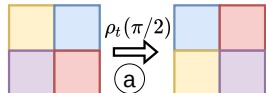 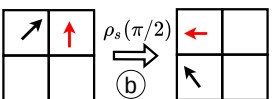 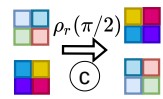

Figure 2: Illustration of a pixel-wise rotation (characterized by a *fixed* representation $\rho_f$) and a channel-wise rotation (characterized by the representation $\rho$). When $g$ is a $\pi/2$ CCW rotation, $\rho_f$ *always* rotates the pixels while the effect of $\rho$ varies, e.g., the effect when $\rho$: **(a)** being a trivial representation ($\rho_t$) acting on a 1-channel feature map, **(b)** being a standard representation ($\rho_s$) acting on a vector field, and **(c)** being a regular representation ($\rho_r$) acting on a 2-channel feature map.

## 3.2 $C_n$ and SO(2) Symmetry Groups

In this work, we are mainly concerned with the symmetry group $G = \text{SO}(2)$ of continuous planar rotation, defined as $\text{SO}(2) = \text{Rot}_\theta : \{0 \leq \theta < 2\pi\}$. For a reduced computation complexity, we use the cyclic subgroup $C_n \leq \text{SO}(2)$ to approximate SO(2), which is defined as $C_n = \{\text{Rot}_\theta : \theta \in \{\frac{2\pi i}{n} \mid 0 \leq i < n\}\}$. In other words, $C_n$ defines $n$ rotations (i.e., group elements), which are multiples of $\frac{2\pi}{n}$. For instance, $C_4 = \{0, \pi/2, 2\pi/2, 3\pi/2\}$ and $C_8 = \{0, \pi/8, \ldots, 6\pi/8, 7\pi/8\}$.

## 3.3 Group Representations

A *group representation* is a mapping from a group $G$ to a $d$-dimensional general linear (GL) group, i.e., $\rho : G \to \text{GL}_d$ by assigning each group element $g \in G$ with an invertible matrix $\rho(g) \in \mathbb{R}^{d \times d}$.

When $G = C_n$, the effect of a rotation $g \in C_n$ on a signal $x$ (i.e., $gx$) starts with a *pixel-wise* rotation $\rho_f(g)^{-1}x$ (with a *fixed* group representation $\rho_f$), followed by a *channel-wise* rotation, i.e., $gx = \rho(g)(\rho_f(g)^{-1}x)$ (with the choice of group representation $\rho$). In this work, we consider three choices of the channel-wise representation $\rho$:

**Trivial Representation ($\rho = \rho_t$):** For $\forall g \in G$, $\rho_t$ associates $g$ with an identity matrix. For example, in Fig. 2a when $g$ is a $\pi/2$ counter-clockwise (CCW) rotation, and $x$ is a 1-channel feature map, $\rho_f$ rotates the pixels of $x$ while $\rho_t$ does not change the pixel values (i.e., the colors are unchanged).

**Standard Representation ($\rho = \rho_s$):** For $\forall g \in G$, $\rho_s$ associates $g$ with a rotational matrix, i.e., $\rho_s(g) = g$. As in Fig. 2b, when $g$ is a $\pi/2$ CCW rotation and $x$ is a vector field input, $\rho_f$ rotates the positions of vectors (denoted as colored arrows), and $\rho_s$ rotates their orientations.

**Regular Representation ($\rho = \rho_r$):** For each $g \in G$, when acting on an input $x$, $p_r$ will cyclically permute the coordinates of $x$. Fig. 2c illustrates when $g$ is a $\pi/2$ CCW rotation and $x$ is a 2-channel feature map, $\rho_f$ rotates each channel's pixels and $\rho_r$ permutes the orders of the channels.

**An Illustrative Example** Combining the group and the group representation fully characterizes how a signal will be transformed. For an illustrative example in a grid-world domain, see Appendix A.

## 3.4 Equivariance, Invariance, and Group-invariant MDPs

Given $\phi\colon \mathcal{X} \to \mathcal{Y}$ and a symmetric group $G$ that acts on $\mathcal{X}$ and $\mathcal{Y}$, we say that $\phi$ is $G$-*equivariant* if $\phi(gx) = g\phi(x)$, and $G$-*invariant* if $\phi(gx) = \phi(x)$. For the remainder of this document, we drop the prefix $G$ and simply refer to these properties as invariance and equivariance.

These notions have been adopted in the framework of group-invariant MDPs [1]. Specifically, an MDP $M_G = (\mathcal{S}, \mathcal{A}, T, R)$ is invariant if the transition and the reward function are invariant, i.e., $T(gs, ga, gs') = T(s, a, s')$ and $R(gs, ga) = R(s, a)$. Group-invariant MDPs are associated with an invariant optimal Q-function, i.e., $Q^*(gs, ga) = Q^*(s, a)$, and at least one equivariant deterministic optimal policy, i.e., $\pi^*(gs) = g\pi^*(s)$. These properties were exploited to build very sample-efficient equivariant agents under full observability [2, 1, 5, 3].

# 4 Group-Invariant POMDPs

In this section, we extend the ideas from [1] to POMDPs and identify the basic set of assumptions that a POMDP needs to satisfy to have analogous invariance properties. We also note that while other assumptions might also lead to an invariant POMDP, ours are probably the most natural.

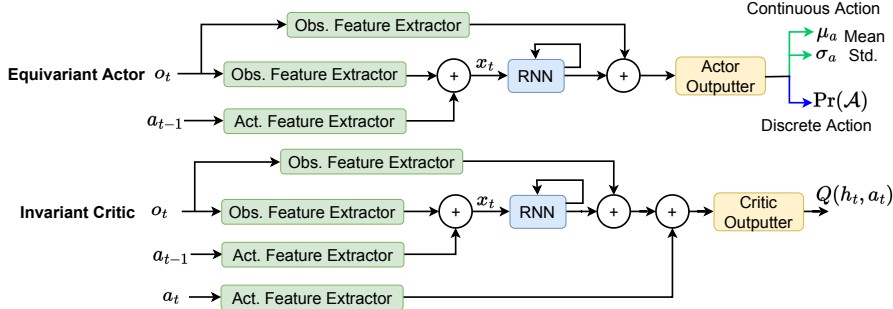

Figure 3: Our equivariant agent takes the commonly used structure of a memory-based actor-critic agent [13, 42, 43, 44] but consists of an equivariant actor and an invariant critic, each constructed by equivariant modules. The actor's output can be learned means and standard deviations (for continuous action spaces) or a categorical distribution over the action space (for discrete action spaces).

**Definition 1.** *We say a POMDP $P_G = (\mathcal{S}, \mathcal{A}, \Omega, b_0, T, R, O)$ is group-invariant with respect to group $G$ if it satisfies the following invariant properties for all $g \in G$:*

$$
\begin{aligned}
T(gs, ga, gs') = T(s, a, s') \quad & R(gs, ga) = R(s, a) \\
O(ga, gs', go) = O(a, s', o) \quad & b_0(gs) = b_0(s) \,.
\end{aligned}
\tag{1}
$$

This extends the definition of the group invariant MDP from [1] by incorporating additional constraints on the observation function and the initial belief distribution. We also extend the group operations on histories.

**Definition 2.** *Group operation $g$ acts on history $h_t$ according to $gh_t := (go_0, ga_0, \ldots, ga_{t-1}, go_t)$.*

Finally, we show that group-invariant POMDPs exhibit similar properties and benefits as group-invariant MDPs.

**Theorem 1.** *A group-invariant POMDP has an invariant optimal $Q$-function $Q^*(gh, ga) = Q^*(h, a)$, an invariant optimal value function $V^*(gh) = V^*(h)$, and at least one equivariant deterministic optimal policy $\pi^*(gh) = g\pi^*(h)$.*

*Proof.* See Appendix B. □

The above analysis allows us to constrain the value function and policy for a $G$-invariant POMDP to be invariant and equivariant, respectively, without eliminating optimal solutions.

## 5 Equivariant Actor-Critic RL for POMDPs

In this section, we introduce an equivariant agent that directly exhibits the desired properties of the optimal value function and policy, backed by the analysis in Theorem 1. Fig. 3 shows the agent, which takes a typical memory-based agent [13, 42, 43, 44] but has an equivariant actor and an invariant critic, each consisting of equivariant models. We later show that this very generic agent, when embedded with the domain symmetry, can outperform significantly strong POMDP methods.

### 5.1 Equivariant Modules

We describe the details of the equivariant modules within our agent below, with the core components being *equivariant CNNs* [45, 32]. For the implementation details, please see Appendix D.

**Equivariant Feature Extractor** This module takes observations or actions and outputs intermediate features for further processing. It comprises multiple equivariant CNN components chained sequentially as shown in Fig. 4a. Its input representation is the observation representation $\rho_o$ or the action representation $\rho_a$. The intermediate and output representations are chosen to be the regular representation $\rho_r$, which empirically outperforms other representations [32]. The input representation can be a single representation type or *mixed*, i.e., a sum of different representations. A mixed representation is necessary when the input has different components that transform differently under a group

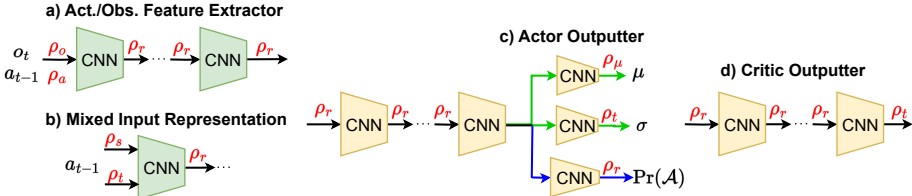

Figure 4: Equivariant Feature Extractor and Actor/Critic Outputter modules.

transformation, e.g., one component rotates with the transform, and one component is unchanged. Such a case can be seen in Appendix D.2 and is simplified in Fig. 4b, where $\rho_a = \rho_s + \rho_t$.

**Equivariant Actor Outputter** Fig. 4c shows the input representation is regular because the signal coming in (the output of the RNN and the observation feature extractor modules) uses a regular representation. The output representation varies depending on the action type (discrete/continuous) and how a group transformation will affect an action. For *discrete actions*, the module produces a categorical distribution over the action space. In this case, the output representation is the regular one $\rho_r$ as we want to change the (discrete) action when the history is transformed (see Appendix A for an illustration). For *continuous actions*, this module outputs the means with some representation $\rho_\mu$ and the standard deviations of actions with some representation $\rho_\sigma$ (as in A2C [7], PPO [46], or SAC [8]). The representations used for $\rho_\mu$ and $\rho_\sigma$ are mixed, as each action component might change differently under a group transformation (see Appendix D.2).

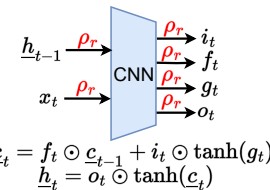

$$\underline{c}_t = f_t \odot \underline{c}_{t-1} + i_t \odot \tanh(g_t)$$
$$\underline{h}_t = o_t \odot \tanh(\underline{c}_t)$$

Figure 5: Equi. LSTM cell.

**Equivariant Critic Outputter** Because the optimal critic is invariant, this module (Fig. 4d) uses the trivial representation $\rho_t$ at the output to keep the output the same under a group transformation. Its input representation is regular, enforced by the output of the RNN and the action feature extractor.

**Equivariant Recurrent Neural Network** This is our contribution needed for constructing a POMDP equivariant agent (there is another similar component in [26], but only model weights are released). We utilize an LSTM [47] for this module, but the approach can also be modified for other types of RNNs. Specifically, given an input $x_t$ (e.g., the concatenated obs-action feature) and the previous hidden state $\underline{h}_{t-1}$, the input gate $i_t$, the forget gate $f_t$, the memory cell candidate $g_t$, and the output $o_t$ are computed as follows with $W$s and $b$s being learnable weights and biases:

$$i_t = \text{sigmoid}(W_{xi}x_t + W_{hi}\underline{h}_{t-1} + b_i) \quad f_t = \text{sigmoid}(W_{xf}x_t + W_{hf}\underline{h}_{t-1} + b_f)$$
$$o_t = \text{sigmoid}(W_{xo}x_t + W_{ho}\underline{h}_{t-1} + b_o) \quad g_t = \tanh(W_{xg}x_t + W_{hg}\underline{h}_{t-1} + b_g). \tag{2}$$

The above equations do not make an equivariant RNN module. To enforce the equivariance, we compute all equations at once using an equivariant CNN module (Fig. 5), similar to the ConvLSTM network [48]. The input representation $\rho_r$ is determined by the output of the feature extractors ($x_t$) and the previous hidden state ($\underline{h}_{t-1}$), and the output representation is also regular. Next, we compute the next hidden state $\underline{h}_t$ and cell state $\underline{c}_t$ using the common LSTM equations with $\odot$ denoting the Hadamard product:

$$\underline{c}_t = f_t \odot \underline{c}_{t-1} + i_t \odot \tanh(g_t) \quad \underline{h}_t = o_t \odot \tanh(\underline{c}_t). \tag{3}$$

Finally, as the output of the RNN is an approximation of the belief state, to satisfy the condition of an invariant initial belief distribution, we set $\underline{c}_0$ and $\underline{h}_0$ with zero vectors.

## 6 Experiments

We compare the performance of learning agents on two grid-world domains (discrete actions and feature-based observations) and four robot domains (continuous actions and pixel observations).

### 6.1 Domains

We briefly describe our domains below. Please refer to Appendix C for more specific details.

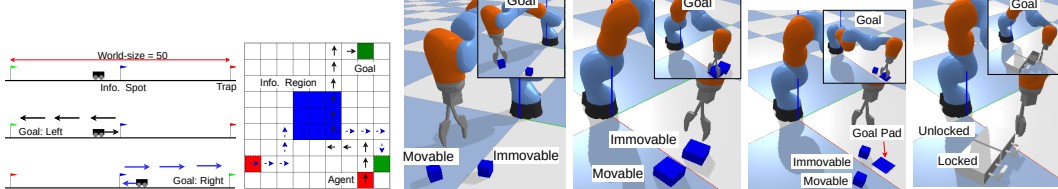

CarFlag-1D    CarFlag-2D    Block-Picking  Block-Pulling  Block-Pushing  Drawer-Opening

Figure 6: Our domains. The first two domains have feature-based observation and discrete action spaces. The last four domains have pixel-based observations and continuous action spaces.

### 6.1.1 Grid-World Domains

There are two versions of CarFlag [49] in Fig. 6, where an agent must reach a goal (green), whose position is visible *only* when the agent visits an unknown information region. For instance, in CarFlag-1D, the agent must visit the central blue flag to get the side (left/right) of the goal; or in CarFlag-2D, the agent must visit the central blue region to see the coordinate of the goal cell. We also illustrate the domain symmetry in the figure: in these domains, when the starting position and the goal location are transformed (flipped in CarFlag-1D or rotated by $\pi/2$ radians clockwise in CarFlag-2D, the optimal trajectories will be transformed similarly, i.e., black $\rightarrow$ blue trajectories.

### 6.1.2 Robot Manipulation Domains

Fig. 6 shows our robot manipulation domains (extended from the BulletArm suite [50]), where a robot arm must perform individual manipulation tasks (i.e., picking, pulling, pushing, and opening) using *top-down* depth images to win a sparse reward. In these domains, only one object is manipulable, but both objects are the same if only relying on the current image. Therefore, the agent must actively check the objects' mobility and remember past interactions with the objects to determine the next action. Specifically, in Block-Picking, the agent needs to pick the movable block up. In Block-Pulling, the agent needs to pull the movable block to be in contact with the other block. In Block-Pushing, the goal is to push the movable block to a goal pad. In Drawer-Opening, the agent is tasked to open an unlocked drawer between a locked and an unlocked one.

In these domains, the transition function is invariant because the Newtonian physics applied to the interaction is invariant to the location of the reference frame. The reward function is invariant by definition. Using top-down depth images makes the observation function invariant. If the initial belief is assumed invariant, then according to Definition 1, these domains are group-invariant POMDPs.

## 6.2 Agents

We compare our proposed agents (instances of the structure in Fig. 3 applied to A2C [7] and SAC [8]) against a diverse set of baselines, including on-policy/off-policy, model-based/model-free, and generic/specialized POMDP methods (see Appendix D and Appendix E for more details).

### 6.2.1 Grid-world Domains

**RA2C** [51] is a recurrent version of A2C [7]. **Equi-RA2C** is our proposed architecture applied to A2C. **DPFRL** [14] is a state-of-the-art model-based POMDP baseline where an A2C agent is given features produced by a differentiable particle filter. **DreamerV2** [52] and **DreamerV3** [53] are strong model-based methods that learn a recurrent world model, thus, can work with POMDPs.

**No Data Augmentation for All** Since all methods are on-policy algorithms , augmented data using the domain symmetry only becomes on-policy *only* for **Equi-RA2C** dues to its unique symmetry-awareness. Therefore, for a fair comparison, we do not perform any data augmentation.

### 6.2.2 Robot Manipulation Domains

While on-policy RA2C or Dreamer-v2 can handle continuous action spaces in these domains, there is no clear way to leverage expert demonstrations necessary to efficiently solve the robot manipulation tasks with sparse rewards in Fig. 6. Thus, we switch to SAC [8] as the base RL algorithm, where

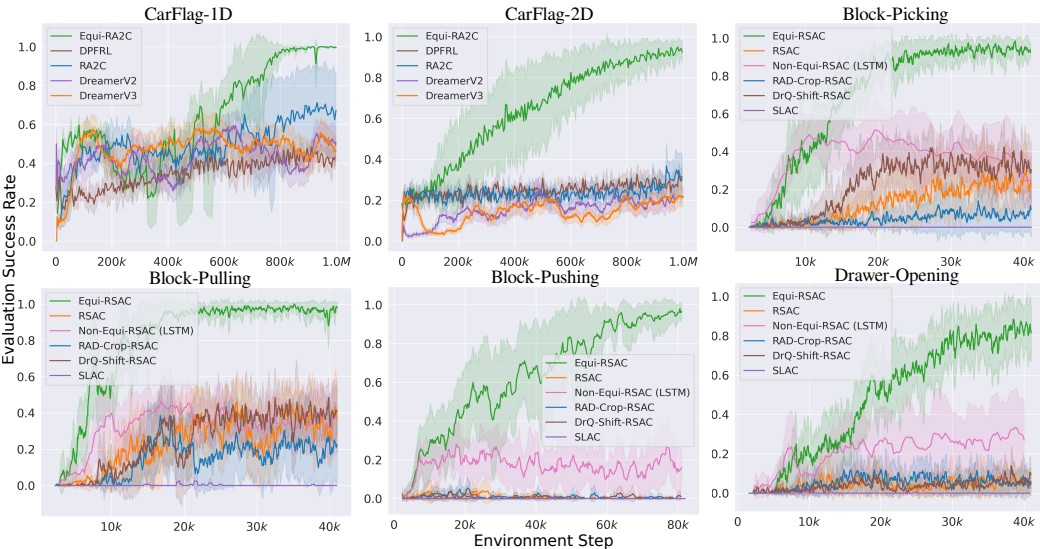

Figure 7: Evaluated success rates (four seeds, shaded areas denote one std.). No data augmentation is used in `CarFlag` domains. Rotational data augmentation is used for *all* agents in robot domains.

we can pre-populate its replay buffer with demonstration episodes. In our experiments, **RSAC** [13] is a non-equivariant recurrent SAC agent. **Equi-RSAC** is our proposed method applied to SAC. We also compare with *recurrent* versions of two strong data augmentation baselines: **RAD-Crop-RSAC** [54] and **DrQ-Shift-RSAC** [55]. These specific data augmentation techniques, i.e., random cropping and shifting (see Appendix G for visualizations), are chosen among others because they were reported to perform best [55]. To train RAD-Crop-RSAC, for each training episode, an auxiliary episode is created by using the same random cropping for every depth image inside the original episode. DrQ-Shift-RSAC applies two random shifts to each depth image in a training episode to create two. The Q-target and the Q-values are then computed by averaging the values computed on the two episodes. Finally, **SLAC** [56] learns a latent model from pixels and then uses SAC on the latent space by using the observation-action history (instead of the latent state) for the actor and the latent state samples for the critic. This enables SLAC to scale to more difficult tasks.

**Demonstrations + Rotational Data Augmentation for All** All replay buffers are pre-populated with 80 expert episodes to overcome the reward sparsity. Moreover, we augment the training data by applying the same random rotation for every action and observation inside a training episode (see Appendix G for visualizations). Note that these rotational data augmentations are applied in addition to the existing data augmentation techniques in RAD-Crop-RSAC and DrQ-Shift-RSAC.

## 6.3 Results

**Grid-world Domains** Fig. 7 shows that Equi-RA2C is significantly more sample efficient than the baselines. Moreover, the dominance of our method is also seen with variants of these domains with different sizes (see Appendix I). DPFRL did not perform well potentially because of the reward sparsity, which was also previously reported in [57]. DreamerV2 and DreamerV3 also perform poorly, even with many more learnable parameters of the models, potentially indicating that learning a good model under partial observability and sparse rewards might be more challenging than in the domains originally tested. For instance, most Atari games and locomotion tasks in the DeepMind Control suite [58] have low levels of partial observability and provide dense rewards.

| $d_{1D}$ | **Equi-RA2C 1M (↑)** | **RA2C 1M (↑)** | $d_{2D}$ | **Equi-RA2C 2M (↑)** | **RA2C 2M (↑)** |
|---|---|---|---|---|---|
| -10 | $0.51 \pm 0.06$ | $\mathbf{0.78} \pm 0.08$ | -2 | $\mathbf{0.38} \pm 0.11$ | $\mathbf{0.31} \pm 0.04$ |
| 10 | $0.44 \pm 0.07$ | $\mathbf{0.70} \pm 0.22$ | 2 | $\mathbf{0.41} \pm 0.12$ | $0.28 \pm 0.02$ |
| -5 | $\mathbf{0.95} \pm 0.03$ | $0.72 \pm 0.33$ | -1 | $\mathbf{0.58} \pm 0.24$ | $0.38 \pm 0.13$ |
| 5 | $\mathbf{0.99} \pm 0.03$ | $0.76 \pm 0.24$ | 1 | $\mathbf{0.71} \pm 0.14$ | $0.30 \pm 0.03$ |

Table 1: The convergent success rates (mean ± one standard deviation) of Equi-RA2C and RA2C agents in asymmetric variants of `CarFlag` (after 1M and 2M training steps). $d_{1D}$ and $d_{2D}$ refer to the distance from the information region to the world center (see Appendix C for illustrations).

**Robot Manipulation Domains** Clearly from Fig. 7, Equi-RSAC strongly outperforms other baselines in all domains, with itself being the only agent that can reach a satisfactory performance. Across all domains, without the equivariant LSTM module (denoted as Non-Equi-RSAC (LSTM), which can be considered as a naive extension of [1]), the performance degrades significantly, even though it starts pretty well. SLAC, surprisingly, performs the worst. A possible reason is that SLAC was originally only tested on domains with dense rewards and low levels of partial observability (e.g., locomotion domains in DeepMind Control Suite [58] and OpenAI Gym [59]). Another potential reason is the usage of concatenated feature vectors across an episode for the actor, which can be very high-dimensional for a long episode. Moreover, we also found that the trained latent model failed to sufficiently reconstruct the observation in `Block-Pulling` (see Appendix F for more details).

**Additional Results** See Appendix H for the performance when using a different group symmetry ($C_8$ instead of $C_4$), utilizing symmetry partially (for either actor or critic only), and $\underline{c}_0$ and $\underline{h}_0$ being random instead of zero vectors. Other additional results are shown in Appendix I.

### 6.4 Using Equivariant Models on Domains with Imperfect Symmetry

We investigate the performance when the perfect symmetry does not hold in asymmetric variants of `CarFlag`, created by offsetting the information region a distance $d$ from the world center (see Appendix C). From the final success rates shown in Table 1 (see Appendix I.1 for learning curves), we can see that equivariant Equi-RA2C still outperforms non-equivariant RA2C when the domains are close to perfect symmetry, i.e., when $d = d_{1D} = 5$ in `CarFlag-1D` or $d = d_{2D} = 1, 2$ in `CarFlag-2D`. However, a bigger symmetry gap might lead to the sub-optimality of equivariant agents. As evidence, Equi-RA2C performs worse than RA2C when $d = 10, -10$ in `CarFlag-1D`.

### 6.5 Zero-shot Transfers to Real Hardware

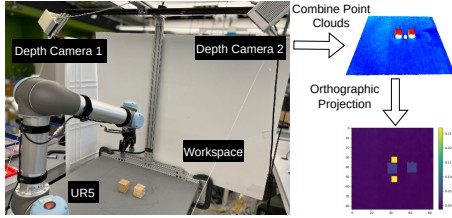

Figure 8: Experimental robot setup.

| Domain | Success Sim / Real ($\uparrow$) |
|---|---|
| `Block-Picking` | 1.00 / 0.90 |
| `Block-Pulling` | 1.00 / 0.88 |
| `Block-Pushing` | 0.96 / 0.92 |
| `Drawer-Opening` | 0.95 / 0.80 |

Table 2: Average success rates of sim2real transfers over 50 episodes.

Because only our agents can perform well in simulation, we transfer their best policies in simulation to a UR5 robot (see Fig. 8). We combine the point clouds from two side-view cameras to create a top-down depth image using a projection at the gripper's position. We roll out 50 episodes, divided equally into test cases when the agents first manipulate the immovable or movable objects. Table 2 shows that the learned policies can be zero-shot transferred well in the real world regardless of small performance drops in all domains (see our supplementary video for policy visualizations). The biggest performance drop is in `Draw-Opening`, in which the transferred policies sometimes clumsily move one drawer far away from the other, creating a novel scene never seen in simulation.

## 7 Conclusion and Limitations

**Conclusion** In this work, we introduced group-invariant POMDPs and proposed equivariant actor-critic RL agents as an effective solution method. Through extensive experiments, our proposed equivariant agents can tackle realistic and challenging robotic manipulation domains much better than non-equivariant approaches with learned policies zero-shot transferable to a real robot.

**Limitations** A limitation of most equivariant approaches, including ours, is the requirement of imperfect symmetry, which might be present when images are affected by non-symmetric factors, e.g., side view instead of top-down view or asymmetric noises. Fortunately, under full observability, recent empirical [60, 61] and theory work [62] show that an equivariant model can still outperform non-equivariant approaches in many such cases. Together with the results in Section 6.4, our approach might still perform better than unstructured agents even under imperfect symmetry.

**Acknowledgments**

This material is supported by the Army Research Office under award number W911NF20-1-0265; the U.S. Office of Naval Research under award number N00014-19-1-2131; NSF grants 1816382, 1830425, 1724257, and 1724191.

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
