# OpenReview forum: "Equivariant Reinforcement Learning under Partial Observability"
_robot-learning.org/CoRL/2023/Conference — CoRL 2023 Poster_

### Official Review · Reviewer_Pu8X · 2023-07-04

**Confidence:** 3
**Originality:** Good
**Technical Quality:** Good
**Clarity Of Presentation:** Good
**Impact:** 3

**Recommendation:**

Weak Accept: I recommend accepting the paper, but will not argue for my recommendation if the majority of other reviewers have a different opinion.

**Review:**

The idea of the paper is nice, the overall presentation seems solid, and the paper is easy to follow. The experiments seem good and even include transfer to real hardware.

I only have some minor points:
- I am very familiar with these POMDP tasks, but for me the sidenote in line 232, that expert demonstrations are necessary in order to solve the task, was a bit surprising. Yes, the tasks is of course much harder, but is Dreamer really not able to solve it at all without expert demonstrations? I think providing a learning curve for dreamer on these tasks would really improve the claim.
- evaluation on four seeds is really not that many.
- I think the presentation of Figure 7 could be quite a bit improved. As of now it is a bit messy:
  - bottom left environment is missing the title
  - move the legends out out the plots and use a shared legend among the different plots. Right now it's a but confusing at first sight, especially since colors are reused between different environments for different algorithms.
  - Pulling out the env names into titles for each plot would also look more organized.
- I believe aligning Figures 2,3,4 at the top of the corresponding pages looks nicer and does not break the flow of text as much.

--------------

After rebuttal

--------------

I thank the reviewers for their response and the effort they put into it. I will keep my current score of weak accept for the paper.

**Quality Of The Limitations Section:**

Limitations are addressed clearly

**Questions For Rebuttal:**

see above.

**Robotics Focus:**

Sufficient demonstration on hardware

**Summary Of Paper:**

The paper extends group invariant MDPs to the POMDP case. It introduces an equivariant LSTM cell and an actor critic agent based on that architecture, that can efficiently learn to solve POMDPs by leveraging symmetries in the environment. The authors experimentally show that their algorithm outperforms baselines and zero-shot transfer the learned policies to a real robot.

**Summary Of Recommendation:**

The paper is well-written and easy to follow. The experiments seem solid and show substantial improvement over the provided baselines. Overall I have very view minor notes on improving the presentation.

---

### Official Review · Reviewer_78Fy · 2023-07-09

**Confidence:** 3
**Originality:** Fair
**Technical Quality:** Very Good
**Clarity Of Presentation:** Good
**Impact:** 3

**Recommendation:**

Weak Accept: I recommend accepting the paper, but will not argue for my recommendation if the majority of other reviewers have a different opinion.

**Review:**

The paper is well-written and easy to read. The authors also provide extensive ablation in the appendix. This paper provides a missing piece in the field of equivariant RL. The experiments are well chosen, and the results are promising.

### Strengths
- well-written and easy to read.
- provides a missing piece in the equivariant RL framework.
- interesting experiments with promising results.


### Weaknesses
- the requirement of symmetries is often very limiting in robotics.
- the question of why symmetries are particularly important in POMDPs is not answered.
- the experiments with a huge number of expert demonstrations limit the results' impressiveness.
- some questions remain.

### Minor Corrections
- line 108, p instead of $\rho$
- Fig 7: no label on the lower left plot; I assume these are the Block-Pulling examples

**Quality Of The Limitations Section:**

Additional details required

**Questions For Rebuttal:**

- why are symmetries particularly important in POMDP?
- how do the agents perform under dense reward signals?
- does the decrease in performance of other agents come from sparse rewards?
- to my understanding, invariant POMDPs require invariant hidden state transition functions, similar to the environment dynamics. Are the dynamics of the hidden transition -- shown in Equations (2) and (3) -- really invariant? Additional clarification is required here.
- did you use convolutional layers for feature extraction for RSAC (and the other approaches)?



**Robotics Focus:**

Sufficient demonstration on hardware

**Summary Of Paper:**

This paper extends the framework of $SO(2)$- Equivariant RL from [1]  to the POMDP setting. Therefore,  the theory is extended to cope with histories instead of states, and an equivariant LSTM structure is proposed. The paper shows strong results in various tasks ranging from simple grid-world tasks to robotic manipulation tasks.



[1] D. Wang, R. Walters, and R. Platt. SO(2) equivariant reinforcement learning. In International
Conference on Learning Representations (ICLR), 2022.


**Summary Of Recommendation:**

I believe that this work makes a notable contribution to the framework of equivariant RL. Hence, I vote for acceptance, but I require my questions to be answered.

---

### Official Review · Reviewer_e6B6 · 2023-07-18

**Confidence:** 3
**Originality:** Very Good
**Technical Quality:** Good
**Clarity Of Presentation:** Very Good
**Impact:** 3

**Recommendation:**

Weak Accept: I recommend accepting the paper, but will not argue for my recommendation if the majority of other reviewers have a different opinion.

**Review:**

Strengths:
- This work introduces an equivariant reinforcement learning method for POMDPs, which is a novel approach to the best of my knowledge.
- The problem and solution are presented very clearly, and the visualizations are effective in illustrating the method components.
- The results are very strong and demonstrate the advantage of encoding symmetry into the agent, both theoretically and empirically in several POMDP settings.

Weaknesses:
- The condition under which this method excels is when there is perfect or near-perfect symmetry in the problem. The symmetry must be known beforehand in order to encode this symmetry into each component network of the agent.
- The experimental analysis is quite thorough. Perhaps, the main missing component is a comparison to POMDP methods that leverage privileged information about the task at train time, e.g., of the hidden state, a task identifier. The problems studied in the experiments can also be viewed as meta-RL tasks, so I wonder how a meta-RL method such as DREAM [1] would perform on them.

[1] Liu et al. Decoupling Exploration and Exploitation for Meta-Reinforcement Learning without Sacrifices. ICML 2021.

**Quality Of The Limitations Section:**

Limitations are addressed clearly

**Questions For Rebuttal:**

- Why is data augmentation not applied to the on-policy algorithms? (Lines 228-229)

- Why do you think the data augmentation methods plateau at such low success rates? It seems like it may be more than just sample-inefficiency. Even Non-equi-RSAC seems to outperform the variant with data augmentation, which is not obvious why.

- Why does the method take 1M environment steps to learn in the CarFlag environments, while it takes 40-80K steps in the robot manipulation domains? The episodes seem to have comparable lengths.

- How would other POMDP methods that leverage privileged information at train time, e.g., [20-23] referenced in the paper, perform on these tasks?

- Could this method take advantage of symmetry that exists even if there are also asymmetric components in the problem?

**Robotics Focus:**

Sufficient demonstration on hardware

**Summary Of Paper:**

This paper extends equivariant reinforcement learning to POMDPs. Specifically, it shows that in group-invariant POMDPs, the optimal value function must be invariant and the optimal policy must be equivariant. To implement such a solution, this work introduces an equivariant actor-critic method that embeds the domain symmetry into its encoder network, recurrent network, and actor/critic components. The experiments are conducted in feature-based gridworld and a simulated pixel-based robot manipulation suite based on BulletArm. The built-in equivariance significantly improves the performance and sample-efficiency of the learning agent, compared to non-equivariant networks and those trained with rotational data augmentation.

**Summary Of Recommendation:**

This paper studies equivariant RL in symmetric POMDPs, and presents the advantages of doing so both empirically and theoretically. There are a few remaining questions I have in terms of its generality and whether related solutions may be applicable here, but I generally think this paper would be a good fit for CoRL.

---

### Official Review · Reviewer_jfAZ · 2023-07-19

**Confidence:** 4
**Originality:** Good
**Technical Quality:** Good
**Clarity Of Presentation:** Good
**Impact:** 3

**Recommendation:**

Weak Accept: I recommend accepting the paper, but will not argue for my recommendation if the majority of other reviewers have a different opinion.

**Review:**

see please "Questions For Rebuttal"

**Quality Of The Limitations Section:**

Limitations are addressed clearly

**Questions For Rebuttal:**

1. line 102: what does "fiber-wise rotation" mean? is there a formal definition beyond an illustrative example?

2. line 148: ".. memory-based agent..". why is the agent memory-based? How does it follow from Theorem 1? Is it an assumption? Heuristics?

3.  What is you design criteria for the scheme in Figure 3? Is it the only scheme? What are special in this scheme?

4. "We later show that this very general agent," is not a very general agent, but a memory-
based agent

**Robotics Focus:**

Highly relevant to robotics but no hardware experiments

**Summary Of Paper:**

this paper extent equivariant MDP to equivariant POMDP. A theoretical justification and a particular implementation with deep neural networks (recurrent, MLP and transforms) are provided. The method is demonstrated both in simulated and real robots.

**Summary Of Recommendation:**

a complete story is provide: idea, it theoretical justification, and demonstration both on the simulators and a robot.

---

### Author Response · Authors · 2023-08-11
**Summary of Changes**

We highly appreciate all reviewers for suggestions to improve our work. Here is the summary of changes that we did/are doing for the paper:

- We addressed points made by all reviewers and replied separately in the attached files. We also updated the paper accordingly (attached in each zip file in reply to each reviewer)
- Finished experiments show the agents' performance under dense rewards as asked by Reviewer 78Fy. The results still show the substantial advantages of our proposed method to other baselines
- Finished experiments to show that Dreamer-v2 does not learn in the robot manipulation domains without no expert demonstrations, asked by Reviewer Pu8X
- Running experiments w/ more seeds (changing from four to six seeds) as asked by Reviewer Pu8X. We will update the paper when we have all results. We, however, do not expect substantial changes in the results

---

### Decision · Program_Chairs · 2023-08-30

**Decision:**

Accept (Poster)

**Comment:**

The reviewers agree that this paper makes a worthy contribution to CoRL -- it introduces a method that incorporates equivariance into RL agents in partially-observed problems. The paper is written well, and the experimental results are compelling. I encourage the authors to incorporate the reviewer feedback in the final version.